# On-Demand Release of Fucoidan from a Multilayered Nanofiber Patch for the Killing of Oral Squamous Cancer Cells and Promotion of Epithelial Regeneration

**DOI:** 10.3390/jfb13040167

**Published:** 2022-09-28

**Authors:** Yingnan Liu, Yingjie Xu, Xiaopei Zhang, Na Liu, Beibei Cong, Yu Sun, Mingxia Guo, Zeyu Liu, Le Jiang, Wanchun Wang, Tong Wu, Yuanfei Wang

**Affiliations:** 1Institute of Neuroregeneration and Neurorehabilitation, Qingdao Medical College, Qingdao University, Qingdao 266071, China; 2Qingdao Stomatological Hospital, Qingdao 266001, China; 3Shandong Key Laboratory of Medical and Health Textile Materials, Collaborative Innovation Center for Eco-Textiles of Shandong Province and the Ministry of Education, 308 Ningxia Road, Qingdao 266071, China

**Keywords:** electrospun nanofibers, phase-change material, photothermal effect, controlled release, SCC-9 cells, epithelial regeneration

## Abstract

Oral squamous cell carcinoma represents 90% of all oral cancers. Recurrence prevention remains an important prognostic factor in patients with oral squamous cell carcinoma, and the recovery of the oral epithelium post-surgery is still a challenge. Thus, there is an urgent need to develop a smart carrier material to realize the spatiotemporally controlled release of anticancer drugs, instead of multiple oral administrations, for recurrence prevention and promoting the reconstruction of injured epithelial tissues. Here, we developed a multi-layered nanofiber patch capable of the photothermal-triggered release of low-molecular-weight fucoidan (LMWF) from the sandwiched layer, together with electrospun fibers as the backing and top layers. The sandwiched layer was made of phase-change materials loaded with indocyanine green, a photosensitive dye, for the localized release of LMWF in response to near-infrared irradiation. We showed that the on-demand release of LMWF was able to kill oral cancer cells effectively. Furthermore, adding acellular dermal matrix to the top nanofiber layer improved the proliferation of human oral keratinocytes, while the hydrophobic back layer served as a barrier to prevent loss of the drug. Taken together, this study provides a feasible and smart material system for killing oral squamous cancer cells together with the recovery of oral epithelium.

## 1. Introduction

Oral squamous cell carcinoma, the main pathological type of oral cancer, accounts for 90% of oral cancer [1,2]. The current treatment strategies for oral cancer mainly involve radiotherapy, chemotherapy, modern surgery techniques, immunity therapy, and anticancer drugs. Nevertheless, these methods still have serious side effects and a low survival rate [3,4]. In particular, such strategies were not able to thoroughly kill the cancer cells. This is mainly because of the complex anatomical structure, rich blood supply, and extensive small branches of blood vessels in the maxillofacial region. On the other hand, lymphatic metastasis and postoperative recurrence may also contribute to poor prognosis [5,6]. As such, there is an urgent need to develop new treatment approaches for continuously killing the residual cancer cells after clinic treatment.

The rapid development of biomedical materials highlights their broad application and the promising potential in curing oral cancer. To date, many implantable and degradable biomaterials have exhibited the capabilities of killing cancer cells and inhibiting tumor recurrence due to the enhanced targeting and specificity [7,8,9]. For example, biocompatible and implantable hydrogels, microspheres, aerogels, and electrospun nanofibers have shown good performance in targeting the treatment of breast cancer, colon cancer, and glioma [10,11]. Among others, electrospun nanofibers have been considered a class of promising biomaterials for local cancer therapy because of their high drug loading efficiency, porous nanostructure, and the biomimetics of native extracellular matrix.

In addition to electrospun nanofibers, a variety of gate-keeper materials have also been applied to design intelligent drug delivery platforms in response to specific stimuli-responsive microenvironments (e.g., pH or temperature change) of the lesion to enhance cell targeting specificity and avoid premature release of drugs [10,12,13,14,15,16,17,18]. In these cases, the stimulus-responsive biomaterials would change their confirmation and release drugs into the defects when exposed to certain stimulus conditions. Despite these improvements, realizing spatiotemporally controlled and on-demand release of the payloads remains challenging to improve the treatment efficiency [19]. To solve this problem, various stimuli-responsive drug delivery systems have been developed to respond to external stimulation conditions such as ultrasound, light, and temperature [20,21,22,23,24,25,26]. Among others, the triggered release of payloads using near-infrared (NIR) light has attracted wide attentions because of its profound penetration ability, high spatial and temporal resolutions, and lack of toxicity to cells and tissues [22,27]. Biomaterials integrated with carriers that could photothermally respond to the NIR irradiation would be an ideal system for realizing on-demand release for cancer treatment and tissue regeneration. Among different NIR-responsive biomaterials, phase-change materials (PCMs) have made significant progress in delivering drugs or growth factors in response to the thermal condition, in which the PCMs would change from solid to liquid phase when the ambient temperature is higher than their melting point [28,29,30,31,32,33,34,35,36,37]. Owing to this unique feature, PCMs loaded with drugs and photothermal initiators have been applied to the controlled release of drug or growth factors for accelerating cell migration, promoting neurite outgrowth, and realizing cancer treatment. Previously, our group reported the development of thermo-sensitive PCMs composed of a certain proportion of lauric acid and stearic acid [37]. Such PCMs have a sharp melting point at 39 °C, which would not damage any cells and tissues. For example, Au nanocages containing PCMs and 2,2′-azobis[2-(2-imidazolin-2-yl) propane] dihydrochloride were invented for in situ generating reactive oxygen species under appropriate NIR irradiation [34]. In another case, nerve growth factors and indocyanine green (ICG) were integrated with electrospray PCM microparticles to control the growth factors [35]. Post 808-nm NIR irradiation, the growth factors were released from PCM particles to promote the neurite extension of PC12 cells. Thus, such PCMs are suitable for manufacturing biomedical materials for tissue repair and/or cancer therapy.

Low-molecular-weight fucoidan (LMWF), one of the most critical marine active ingredients, has also been widely investigated for cancer treatment. Previous studies showed that the bioactivity of fucoidan varied according to its molecular weight: low molecular weight (<10 kDa); medium molecular weight (10–10,000 kDa); high molecular weight (>10,000 kDa) [38,39]. In most cases, LMWF has better biological effects, including anti-inflammatory, anti-tumor, antibacterial, antioxidant, and immunomodulatory capability, compared to the other fucoidan species [38,39,40]. In particular, it has been documented that LMWF has a unique feature of reducing the production of reactive oxygen species, affecting the expression of matrix metallopeptidase 2, and inhibiting the growth and proliferation of cancer cells [41,42,43]. For example, LMWF has been found to induce apoptosis of mouse liver cancer cells [44]. In another study, the proliferation of human lung adenocarcinoma cells (i.e., A549 and SPCA-1) was inhibited after treatment with LMWF [45].

Herein, we report a tri-layered patch that combines electrospun nanofibers and PCM via electrospinning and electrospray techniques for oral carcinoma treatment. Such a patch contains a bottom layer consisting of electrospun nanofibers made of a blend of polycaprolactone (PCL) and acellular dermal matrix (ADM). As one of the most useful bioactive materials, ADM has been spread and applied in neurosurgery, otolaryngology, burn plastic surgery, periodontics, oral mucous medicine, etc. The addition of ADM endows the nanofibers with better hydrophilicity and the capacity to promote the adhesion and proliferation of human oral keratinocytes (HOK). Then, a layer consisting of electrospray PCM microparticles encapsulating LMWF and ICG was deposited onto the PCL/ADM nanofibers to realize the spatially controlled and localized release of the LMWF. The laser irradiation was applied by a source of 808-nm NIR laser. Upon NIR irradiation, ICG would absorb the light and convert it to heat, thus increasing the temperature until the melting of PCM shells to achieve the on-demand release of LMWF for the treatment of OSCC. Finally, a hydrophobic layer made of electrospun PCL nanofibers was further covered on the PCM microparticles to construct the tri-layered scaffold. Such a barrier layer was employed to prevent the locally released LMWF from being washed away due to chewing or saliva flow. Using such a hierarchical and multi-functional scaffold, we showed the treatment of OSCC upon on-demand NIR irradiation and controlled release of LMWF, as well as the increased proliferation of HOK cells.

## 2. Experimental Methods

### 2.1. Materials and Reagents

Low-molecular-weight fucoidan (Mw ≈ 8177 Da) was extracted and provided by the Institute of Oceanology, Chinese Academy of Sciences (Qingdao, China). ICG (Catalog no. YZ-1340009) and 4′,6-diamidino-2-phenylindole (DAPI) (Catalog no. S2110) were purchased from Solarbio (Beijing, China). Rhodamine-B (Catalog no. 83689) was purchased from Sigma-Aldrich (St. Louis, Missouri, USA). Bull serum albumin (BSA) (Catalog no. BS114) was obtained from Biosharp (Anhui, China). Lauric acid (97%) (Catalog no. L0011) and stearic acid (95%) (Catalog no. S0163) were obtained from Tokyo Chemical Industry (Tokyo, Japan). Cell Counting Kit-8 (CCK-8) (Catalog no. C6050) was purchased from NCM Biotech (Suzhou, China). PCL (Mw ≈ 80,000 g mol^−1^) (Catalog no. 440744) was obtained from Sigma-Aldrich (St. Louis, MO, USA). Phalloidin-iFluor 488 (Catalog no. ab176753) and Phalloidin-iFluor 555 (Catalog no. ab176756) were purchased from Abcam (Shanghai, China). Dulbecco’s Modified Eagle’s Medium (DMEM) (Catalog no. 12100) and Antibiotic-Antimycotic (Catalog no. P1400) were purchased from Solarbio (Beijing, China). Fetal bovine serum (FBS) (Catalog no. P30-3302) was purchased from Pan (Adenbach, Germany). Trifluoroethanol (TFE) (Catalog no. T63002) was obtained from Sigma-Aldrich (St. Louis, MO, USA). Human OSCC cell lines SCC-9, and HOK cell lines were reanimated from the cryopreserved cells provided by Qingdao Stomatological Hospital (Qingdao, China). Annexin V-APC/PI apoptosis detection kit (Catalog no. E-CK-A217) was bought from Elabscience Biotechnology (Wuhan, China).

### 2.2. Fabrication and Characterization of the Nanofiber Patch

Fabrication of the PCM Microparticles: The PCM microparticles were fabricated using the electrospray process. The outer PCM solution was prepared by adding the mixture containing lauric acid and stearic acid at a mass ratio of 4:1 into a solution of ethanol and dichloromethane (20:80 by volume). A 0.5 wt% gelatin solution containing ICG was produced as the inner solution. Different types of payloads, Rhodamine B (RB), FITC-BSA, and LMWF, were separately added to the core solution at concentrations of 1, 20, and 4.5 mg/mL. The PCM microparticles loaded with RB, FITC-BSA, and LMWF were obtained by electrospray, respectively, with a high voltage of 20 kV, a flow rate of 3 mL/h (shell) and 1 mL/h (core), and a collecting distance of 15 cm.

Fabrication of the Tri-layered Patch: The tri-layered patch was prepared by electrospraying PCM particles between two layers of electrospun fibers (Figure 1). The specific parameters were shown in Table 1. In detail, PCL electrospun nanofibers were first fabricated as the backing layer by electrospinning the TFE solution containing 10 wt% PCL for 30 min. In this step, the flow rate was 2.0 mL/h, and the high voltage was 15 kV. A collecting distance of 15 cm was applied between the needle and the collector. Subsequently, the PCMs loaded with a blend of LMWF and ICG were fabricated by co-axial electrospray and deposited onto the PCL nanofibers. During the co-axial electrospray process, the outer solution was prepared by adding the mixture containing lauric acid and stearic acid at a mass ratio of 4:1 into a solution of ethanol and dichloromethane (20:80 by volume). A 0.5 wt% gelatin solution containing ICG and LMWF was produced as the inner solution. To determine the optimized ICG concentration before final application, 0, 0.1, 0.2, 0.4, 0.8, and 1.0 mg/mL ICG was separately utilized as the inner solution to measure the temperature increase. During the electrospray process, the outer and inner flow rates were set at 3.0 and 1.0 mL/h, respectively. Under the voltage of 20 kV, the PCMs were pumped out through the dual-capillary needle for 10 min. Finally, a 10 wt% PCL/ADM solution was prepared by adding PCL and ADM into TFE at a mass ratio of 1:1. During the electrospinning process, the obtained mixture was added into a 5-mL injection syringe and pumped out at a flow rate of 2.0 mL/h through a 22-gauge blunt needle under a high voltage of 15 kV. This top layer was fabricated and collected onto the PCM microparticles for 60 min. The photograph of the tri-layered patch was captured by a digital camera. The morphologies of the top layer, backing layer, and PCM microparticles were imaged by a scanning electron microscope (SEM), and the average diameter of the PCL and PCL/ADM fibers was analyzed and recorded by ImageJ.

### 2.3. Photothermal Performance of the Sandwiched PCM Microparticles

The photothermal property of PCM was examined by NIR irradiation. The laser irradiation was applied by a source of 808-nm NIR laser. Firstly, the effects of different concentrations of ICG on the photothermal behavior of PCMs were studied. PCMs with different concentrations of ICG (0, 0.1, 0.2, 0.4, 0.8, 1.0 mg/mL) were fixed at the bottom of 24-well plate (in air) or immersed in 500 μL phosphate-buffered saline (PBS). These samples were irradiated with an 808-nm NIR laser at the density of 2200 mW, and the temperature changes were simultaneously recorded during the irradiation process. After determining the appropriate ICG concentration, we also investigated the photothermal behavior under different laser densities. In this case, the samples were irradiated separately at different laser densities at 1800, 2200, 2600, 3000, and 3400 mW, and the temperature change was recorded as a function of time.

### 2.4. In Vitro Release of LMWF under NIR Irradiation

#### 2.4.1. Release of Rhodamine B from PCM upon Heating

To verify the properties of phase change in response to temperature change, the release of RB encapsulated into the PCM microparticles was studied by continuous and interval heating treatment. For constant heating, the obtained PCMs were soaked in a 50-mL centrifuge tube containing 5 mL PBS solution. Then, the centrifuge tube was placed in a thermostatic water bath (37 °C or 40 °C). At each predesigned time point (4, 8, 12, 16, 20 min), 1 mL solution was collected for measurement, and 1 mL fresh PBS was supplemented into the tube. The absorbance was recorded at 560 nm using a spectrophotometric microplate reader.

For interval heating, the tube containing RB-loaded PCMs was carried out at intervals of 20 min or 30 min, or 60 min between adjacent cycles and immediately cooled below the melting point in an ice bath. After centrifugation for 5 min at 12,000 rpm, 1 mL supernatant was collected and detected at 560 nm. The residual solution was quickly supplemented with 1 mL fresh PBS, followed by another round of heating. The absorbance was recorded as mentioned above.

#### 2.4.2. Release of BSA from PCM under NIR Irradiation

To prove the photothermal-triggered release of the payloads from PCM, BSA and ICG were jointly loaded into PCM using the coaxial electrospray technique. Then, the PCM microparticles were placed at the bottom of the 24-well plate and immersed in 600 μL PBS. A NIR laser triggered the release at a density of 2600 mW for 60 s. Then, 100 μL solution was collected, and 100 μL fresh PBS was supplemented. After 0.5 h or 2 h intervals, the NIR irradiation was applied again. The laser was repeated at least five times to investigate when the release of payloads would stop until the release percentage was close to 100%. The BSA content was determined by BCA assay. At least triplicate samples were investigated.

#### 2.4.3. Release of LMWF from PCM under NIR Irradiation

The DMMB assay investigated the LMWF release study. Firstly, 4 mg DMMB dye was dissolved in 250 mL Milli-Q water. Subsequently, glycine (0.75 g) and sodium chloride (400 mg) were added to the as-obtained solution. After adjusting the pH to 3 with 1 N hydrochloric acid, the DMMB solution was filtered using aseptic filtration membranes (0.22 μm) to remove impurities and stored in a dry, dark environment. The LMWF-loaded PCMs collected on the glass coverslip were fixed on the bottom of the 24-well plate and immersed in 500 μL PBS. Post NIR irradiation, 100 μL solution was collected, and the same volume of fresh PBS solution was supplemented. The NIR irradiation was repeated at least five times after 0.5 h or 3 h intervals, respectively, to determine when the release of payloads would stop. During the measurement process, the collected 100-μL solution was mixed with DMMB solution (1.5 mL), and then the mixture was added to a 96-well plate (100 μL/well) to detect the absorbance at 525 nm. Triplicate samples were tested at each measurement. The release concentrations were confirmed by a calibration curve (0–250 μg/mL).

### 2.5. In Vitro Cell Viability Assay

We first studied the viability of cells cultured on LMWF-loaded PCM microparticles. HOK and SCC-9 cell lines were cultured in a DMEM medium containing 10% fetal bovine serum. In vitro cytotoxicity of PCMs was studied by the CCK-8 cell counting kit. HOK cells were seeded at a concentration of 1 × 10^4^ cells/well in the 24-well plates with different groups: control (-), control (+), PCM (-), PCM-ICG (-), PCM-ICG (+), PCM-ICG-LMWF (-), PCM-ICG-LMWF (+), and free equivalent dosage of LMWF. Here “+” indicates the treatment by laser irradiation at a power density of 2600 mW for 60 s. The cytotoxicity of PCMs to HOK cell lines was evaluated and detected using the CCK-8 method after culturing for 1, 3, and 5 days. Each time, the medium in each well was drawn out, and a fresh medium containing 10% CCK-8 reagent was supplemented. Then, cells were incubated for 3 h, and the samples were allocated to a 96-well plate. A spectrophotometric microplate reader recorded the value of absorbance at 450 nm, and each group’s measurements were repeated three times. The capability of killing cancer cells using PCM-ICG-LMWF (+) was also evaluated by the CCK-8 method. After culturing for 1 day, the groups, including control (+), PCM-ICG (+), and PCM-ICG-LMWF (+), were treated by laser irradiation at a density of 2600 mW for 60 s. The SCC-9 cell viability in all groups was measured after additional culturing for 6 h at 37 °C. Then, the cells were continuously cultured until the next round of laser irradiation and measurement. The laser irradiation treatment was implemented once a day until three days. In addition, the cell viability of SCC-9 and HOK on the multi-layered scaffold was also measured using the same CCK-8 method and processing procedure.

The proliferation of HOK cells on the PCL/ADM membrane was also investigated. The PCL and PCL/ADM membranes were fixed on the bottom of a 24-well plate, and HOK cells were seeded at a concentration of 1 × 10^4^ cells/well. The cells seeded on the wells without membranes were chosen as the control group. The proliferation of HOK cells was investigated using the CCK-8 method after culturing for 1, 3, and 5 days. In addition, the cell morphology of SCC-9 and HOK was separately observed. For the SCC-9 cells, after culturing for 1 day, the groups including control (+), PCM-ICG (+), and PCM-ICG-LMWF (+) were treated by laser irradiation at a density of 2600 mW for 60 s before additional culturing for 6 h at 37 °C. Then, all samples were soaked in 4% paraformaldehyde for 30 min, permeabilized using 0.1% Triton X-100 for 5 min at room temperature, and cleaned with PBS three times. Subsequently, the cells were blocked with 1% BSA for 30 min and washed again with PBS three times. In order to observe the changes in cell morphology more intuitively, the cells’ cytoskeletons were stained with rhodamine-conjugated phalloidin (Invitrogen, Waltham, MA, USA). The rhodamine-conjugated phalloidin dye was mixed with PBS solution containing 1% BSA at the ratio of 1:100 and then added to each well (100 μL per well). After 30 min, the samples were rinsed three times with PBS solution. Subsequently, the DAPI dye was added to each well for 30 min. The different samples were observed using an upright fluorescence microscope. The cell morphology of HOK, culturing for five days, was stained and observed through the same process, except that the rhodamine-conjugated phalloidin dye was replaced by Phalloidin-iFluor 488.

### 2.6. Cell Apoptosis Detection Assay

According to instructions, the SCC-9 cell apoptosis assay was detected by the DxFLEX flow cytometer (Beckman Coulter, Fullerton, CA, USA). The cells were seeded in a 24-well plate at a concentration of 1 × 10^4^ cells/well based on the groups demonstrated in 2.5. After culturing for 1 day, the control (+), PCM-ICG (+), and PCM-ICG-LMWF (+) groups were irradiated by the NIR laser at a density of 2600 mW for 60 s, and the other groups were without irradiation treatment. Three parallel samples were evaluated for each group. The cells were collected by centrifugation at a speed of 900 rpm for 5 min and then rinsed with PBS solution using the same centrifugation process twice. The cells were resuspended with 100 μL diluted 1× Annexin V Binding Buffer solution and then added in 2.5 μL Annexin V-APC and 2.5 μL propidine iodide (PI), followed by incubating in dark conditions at room temperature for 15 min. The results were obtained by the flow cytometer and analyzed with CytExpert for DxFLEX software (Beckman Coulter, Fullerton, CA, USA).

### 2.7. Characterization

The material’s diameter, size distributions, and morphology were observed by scanning electron microscopy (SEM, SUPRA^TM^ 55, Zeiss, Berlin, Germany). The NIR was produced by NIR laser (HW808AD, Hongwaixian, Shenzhen, China). Therm infrared imager recorded the temperature change (Testo869, Detu, Shanghai, China). The release profiles of RB, BSA, and LMWF released from PCM were measured by Micro UV–vis spectrophotometer (NanoDrop™ One, Thermo Fisher, Waltham, MA, USA). The electrospun nanofiber was produced by electrospinning equipment (ET-2535H, Ucalery Technology, Beijing, China). The fluorescence images of PCM nanoparticles loaded with RB were observed by fluorescence microscope (DP74, Olympus, Tokyo, Japan). The cell morphology of HOK and SCC-9 were observed by fluorescence microscope (Nikon DS-Ri2 Zoom, Nikon, Tokyo, Japan). The optical images were obtained by digital cameras (Nikon D7500, Nikon, Tokyo, Japan). The apoptosis data were obtained by a flow cytometer (CytoFLEX, Beckman Coulter, Fullerton, CA, USA).

### 2.8. Statistical Analysis

One-way ANOVA carried out the multiple comparison procedures between different groups. Each group was repeated at least three times, and the statistical data were presented in the way of mean ± standard deviation. A significant difference was determined if *p* < 0.05. The statistical analysis on the temperature change, in vitro cargos release, and cell viability as implemented by origin 2018. The size of nanofiber diameter and nanoparticles were measured by Image J. The apoptosis data from flow cytometry were recorded and processed by CytExpert for DxFLEX 2.0 software. The fluorescence images of PCM nanoparticles loaded with RB were recorded and processed by cellSens Dimension software. The cell morphology of HOK and SCC-9 were recorded and processed by software called NIS-ELEMENTS D.

## 3. Results and Discussion

### 3.1. Fabrication and Characterization of LMWF and the Scaffold

Our previous study proved that low-molecular high-sulfated polysaccharides exhibited cytotoxicity toward SCC-9 cells but little cytotoxicity toward HOK cells [46]. In this study, we first extracted the LMWF and analyzed the composition. LMWF was extracted and produced according to the procedure in a previously published study. Briefly, dry algae were autoclaved in water. The hot solution, separated by continuous filtration, was concentrated and dialyzed. The fucoidan was precipitated and washed three times, then dried at 80 °C. The monosaccharide composition of the LMWF was determined by a pre-column derivatization method. The molecular weight of fucoidan was measured and recorded using high-performance liquid chromatography. The composition analysis and the molecular weight are displayed in Figure 1. The component we wanted was a low-molecular high-sulfated polysaccharide. From Figure 1, we can see that the low-molecular high-sulfated polysaccharides, including sulfate and fucose, constitute the main composition, 36.85%, and 35.07%, respectively. Therefore, we could conclude that the LMWF was extracted successfully as expected.

The multi-layer scaffold containing PCM microparticles was developed by a sequential electrospinning and electrospray process, as shown in Figure 1. As one of the surface layers of the scaffold that would adhere to the damaged area of the oral cavity, the top layer was randomly fabricated by electrospinning PCL and ADM. Many previous studies have reported that PCL is appropriate for cell culture due to its high biocompatibility and mechanical properties. It has been widely applied in fabricating bioactive scaffolds for in vitro cell culturing and in vivo implantation. The addition of ADM enhanced the hydrophilic performance, which improved the mucoadhesive properties of the film. Compared with the water angle of PCL (130.7°), PCL/ADM fibers showed a smaller water angle (44.4°), which was the result of its enhanced hydrophilic performance (Appendix A). Moreover, the enhanced proliferation of HOK cells influenced by ADM further demonstrated its potential for cancer treatment. The PCL/ADM fibers significantly promoted cell viability (Appendix A). The PCL electrospun nanofiber was chosen as the backing layer due to its excellent mechanical strength and hydrophobic properties. The design was intended to prevent the scaffold from falling off during chewing, tongue agitation, and flooding from saliva or other liquids. As shown in Figure 2C,D, the scanning electron microscopy (SEM) images revealed the scaffold details, including the uniaxially random top layer and backing layer, collected on silicon paper. The average diameters of the random PCL fibers and random PCL/ADM fibers were 620.22 ± 161.91 nm and 625.01 ± 139.11 nm, respectively. From Appendix A, we could observe that the diameter distribution of PCL and PCL/ADM nanofibers showed a relatively centralized distribution at approximately 500−600 nm. The PCM microparticles between the two layers were fixed by the coaxial electrospray technique. The fluorescence photograph in Figure 2A was used to determine the payloads encapsulated in PCM particles. The SEM image in Figure 2B shows the distributed microparticles. Although these particles were not ideally uniform in size (589.55 ± 163.94 nm), the size distribution was concentrated on 500–600 nm (Appendix A). The morphology and non-uniform distribution of PCM microparticles did not influence the loading efficiency and release of encapsulated substances. The electrospun nanofibers were deposited directly on a glass coverslip during fabrication. Thus, it was difficult to obtain the thickness of the multi-layer scaffold. The scaffold was estimated to be several micrometers thick by calculating the collection time. The sectional layer of the multi-layer scaffold confirmed the estimated thickness (Appendix A).

### 3.2. Photothermal Properties of the Scaffold

We then studied the photothermal properties of the PCM microparticles obtained by the electrospray process. First, the influence of ICG at different concentrations was investigated. Before the electrospray process, various weights of ICG were added to the gelatin solution as the inner solution. The temperature changes of the microparticles with different concentrations of ICG were recorded in air and PBS. As shown in Figure 3A, the PCM microparticles with higher ICG concentrations exhibited higher plateau temperatures in the air than lower ICG concentrations under the same irradiation density (2200 mW) and time. When the concentration of ICG was in the range of 0.4–1.0 mg/mL, all the temperatures reached 39 °C, the melting point of the PCM. Moreover, the time required to reach the melting point temperature decreased as the ICG concentration increased. Considering the specific physiological environment in which the material would be applied, we also investigated the photothermal behavior of the scaffold in PBS. As shown in Figure 3B, the results in PBS differed from those in air. When the concentration of ICG was 0.4 and 0.8 mg/mL, the terminal temperatures no longer reached 39 °C, and instead plateaued at 33.1 ± 0.2 °C and 37.7 ± 0.2 °C (n = 3), respectively. This phenomenon could be attributed to the existence of PBS raising the thermal conductivity, preventing the temperature from rising. When the concentration of ICG in the PCM microparticles was 1.0 mg/mL, the terminal temperature was again higher than the melting point, 39.2 ± 0.1 °C (n = 3), although it took longer for the PCM microparticles to reach the terminal temperature in PBS than in air. Thus, the 1.0 mg/mL concentration was chosen as the ideal concentration. In addition, the effect of irradiation density on the photothermal behavior of the scaffold was also investigated in air and PBS. It was clear from the results that as the laser density increased, the faster the temperature rose and the shorter the time it took to reach the melting point of the PCM. Based on the results shown in Figure 3C,D, we selected 2600 mW as the ideal laser density because the temperature of irradiated PCM microparticles rose to 39 °C in PBS within 60 s at this laser density. Furthermore, this is an appropriate treatment period for oral cancer therapy without requiring a high laser density. Next, to test the stability of the photothermal behavior, we investigated the temperature change in ICG-loaded PCM microparticles during five on–off cycles in air and PBS. As shown in Figure 4A,B, no significant differences in photothermal behavior were found in the air or PBS after five on–off cycles. In summary, we confirmed that ICG-loaded PCMs could transfer optical energy to thermal energy with stable photothermal performance.

### 3.3. Release of Substances from the PCM Microparticles

Before we investigated the release of LMWF, RB and FITC-BSA were selected as model molecules representing small organic molecules and proteins, respectively, for proof of concept. Encapsulation efficiency was used to measure the extent of the release, which was calculated as the ratio of actual encapsulated content to the theoretical amount. In our study, the encapsulation efficiencies of RB and FITC-BSA loaded in PCM microparticles were 82.68% and 71.81%, respectively. To investigate the thermal-sensitive property of PCM microparticles, we investigated the release of RB encapsulated in PCM microparticles at 37 °C or 40 °C. As shown in Figure 4C, only 2.87 ± 0.07% (n = 3) of RB was detected as being released at 37 °C. We concluded that the slight release of RB was deposited on the surface of the PCM microparticles because the inner solution could not be entirely encapsulated by the outer solution during the electrospray process. In comparison, when RB-loaded PCM microparticles were heated to 40 °C, 96.14 ± 0.15% (n = 3) of the RB was released from the microparticles. Thus, it was suggested that the PCM exhibited sustained release behavior once the temperature was the higher melting point. The release behavior was also studied under heating/cooling cycles to further examine the thermal-responsive behavior. As shown in Figure 4D, RB was released rapidly during heating, but the cumulative release changed very little during the cooling stage. Combining with the results from Figure 4C, we could conclude that the release behavior was temperature sensitive; once the temperature exceeded the PCM melting point, the cargo could be released from microparticles easily. When the ambient temperature was lower than the PCM melting point, hardly any cargo was released from nanoparticles. To summarize, the PCM nanoparticles were controlled-release platforms depending on temperature.

Next, the release behavior of cargos from PCM microparticles under photothermal stimulation was analyzed. ICG, as a NIR absorber, was loaded into the inner solution along with the model molecules, RB and FITC-BSA. We recorded and calculated the cumulative release of each substance after an on–off cycle, and the process was repeated five times. Figure 5A shows the release profile of RB after five cycles of NIR irradiation. We found that the total release of RB was 72.63 ± 1.54%, which was significantly different from the release profile without laser stimulation. Although the cumulative release upon irradiation was lower than that of heating at 40 °C, the total release was still considerable compared with the release behavior in the absence of five cycles of NIR irradiation. The release of RB stopped after ten cycles with an interval of 0.5 h, and the cumulative release of RB had reached 99.13 ± 0.82% (Appendix A). When the irradiation interval was changed from 0.5 h to 2 h, the release profile of RB was 94.25 ± 0.56% after five cycles of NIR irradiation, and the release would stop post eight cycles (99.98 ± 1.60%, Appendix A). We also studied the release of FITC-BSA, the other model bioactive molecule. The release profile shows that 81.45 ± 1.24% of FITC-BSA was released within five cycles of NIR stimulation (Figure 5B). However, only 6.24 ± 0.35% of the payload was detected without laser treatment. The FITC-BSA had stopped the release after ten cycles with an interval of 0.5 h and reached a total release of 97.10 ± 4.75% (Appendix A). When the irradiation interval was changed from 0.5 h to 2 h, the release of FITC-BSA was 91.40 ± 1.00% after five cycles of NIR irradiation. After eight irradiation rounds, the release arrived at 98.65 ± 0.24% and stopped releasing (Appendix A). Thus, we can conclude that PCM microparticles loaded with ICG are promising drug delivery vectors for controlled release in response to NIR stimulation. Subsequently, we measured and recorded the extent of release of the anticancer drug LMWF used in oral cancer treatment. As shown in Figure 5C, the release profile was similar to that of FITC-BSA. The results suggested that the total release of LMWF was 97.54 ± 4.95% after five cycles of NIR irradiation with 0.5 h intervals. The cumulative release of LMWF had reached 98.55 ± 4.41% and stopped releasing after six laser cycles (Appendix A). When the irradiation interval was changed from 0.5 h to 2 h, the release percentage of LMWF reached 97.99 ± 1.82% post five rounds of NIR (Appendix A).

### 3.4. Evaluation of the Treatment Effect on HOK and SCC-9 Cells

Before evaluating the effect of PCM nanoparticles and nanofiber patches, we first investigated the appropriate concentration range of LMWF to use in OSCC therapy. LMWF did not induce cytotoxicity of HOK cells until the concentration reached 250 μg/mL (Appendix A). However, all the LMWF concentrations, ranging from 50 to 1000 μg/mL, promoted apoptosis in SCC-9 cells. (Appendix A). These results were responding to our published works [46]. Therefore, we inferred that the effective therapeutic concentration of LMWF is between 50 and 250 μg/mL and that LMWF is a potential drug for the treatment of OSCC. The extent of release of LMWF was detected after each round of irradiation using the DMMB method, as previously reported. The encapsulation efficiency of LMWF in PCM microparticles was 84.48%. With an increase in the number of NIR stimulations, the cumulative LMWF release gradually increased, and almost all of the LMWF was released after five rounds of irradiation, accounting for 98.5 ± 0.16% of the LMWF. Our previous work indicated that the effective therapeutic concentration of LMWF was between 50 and 250 μg/mL. To achieve this target, an appropriate concentration of LMWF was added to the inner solution based on collection time and flow rate during the process of co-axial electrospray so that the cumulative release of LMWF would achieve a therapeutic concentration at each round of irradiation. We found that the concentration of released LMWF reached 118.13 ± 8 μg/mL after one round of NIR stimulation. After the following two exposures, the concentration of cumulative released LMWF was 159.5 ± 3.2 μg/mL and 201.1 ± 6.5 μg/mL, which was within the therapeutic concentration range. Moreover, the concentration of released drugs was approximately 100 μg/mL or lower in each round. Thus, the drugs released would not affect HOK cell viability. The concentration only reached 7.9 ± 1.4 μg/mL without irradiation. In conclusion, our designed PCMs exhibited NIR-triggered drug release at therapeutic concentrations. In addition, the melting point temperature affected the release of model molecules and LMWF.

To demonstrate therapeutic results in HOK and SCC-9 cells, we quantified the cell viability of these two cell types cultured on PCM microparticles using the CCK-8 method [47]. Whether treated by irradiation or not, there was no significant difference in the HOK cell viability among the test groups. In contrast, the cell survival rates of SCC-9 cells in the PCM-ICG-LMWF (+) and free equivalent dosage of LMWF treatment groups dropped to 43.02 ± 0.86% and 42.68 ± 1.48%, respectively. More importantly, after three treatments, the viability of cells cultured in these two groups was only 4.85 ± 0.42% and 5.02 ± 0.45%, respectively. These experimental results indicate that the designed drug delivery nanosystem exerted the same anti-cancer effect as the direct administration of the same anticancer drugs and did not need to be administrated on time. We also observed that the cell viability in group PCM-ICG-LMWF was 89.96 ± 2.06%, which may be due to the released LMWF collecting on the surface and not contained in PCM microparticles. The cell viability of the other groups remained unchanged (Appendix A). Therefore, we concluded that the melting point temperature (39 °C) reached by a single thermal from NIR stimulation was insufficient to kill cancer cells.

Next, we continued to investigate cells’ viability in a multi-layered nanofiber patch in which PCM microparticles were sandwiched between two layers from different groups. As shown in Figure 6A, HOK cells showed the same trend as single PCM microparticles, except for the control and free equivalent dosage of LMWF groups. Although the absorption of the two groups was lower than that of the other groups, this could be explained by the addition of ADM, which promoted cell proliferation in the other groups. Similarly, the activity of SCC-9 cells exhibited a similar trend compared with cells cultured on PCM particles. Cell viability decreased to 5.06 ± 0.83% in the PCM-ICG-LMWF (+) group after three rounds of irradiation (Figure 7A). Thus, the data indicate that adding a top and back layer would not affect the release behavior of LMWF from PCM microparticles and that the treatment effect was equivalent to delivering the drugs directly (4.82 ± 0.32%). Considering the practical application, the local release of drugs from a multi-layered scaffold could prevent the drugs from being washed away from the oral cavity by saliva, water, or food. Therefore, the scaffold provides flexible drug delivery and release methods with a potentially more significant curative effect for oral applications than direct medicine uptake. The fluorescence images of cell morphology further confirmed these results. In the fluorescence images of HOK cells, we can observe that the cell morphologies in all groups were unchanged in the control group (Figure 6B and Appendix A). The fluorescence images show that the morphology of SCC-9 cells was also almost intact in many groups, including control (-), control (+), PCM (-), PCM-ICG (-), PCM-ICG (+), and PCM-ICG-LMWF (-), which was consistent with the cell viability results. However, cells in the PCM-ICG-LMWF (+) group retained very little normal cell morphology compared with the control group and the PCM-ICG-LMWF (-) group without NIR irradiation (Figure 7B and Appendix A). Therefore, we concluded that the limited release of drugs not embedded in PCMs would not significantly affect cell morphology. In contrast, the release of NIR laser could damage cancer cells morphologies extensively, similar to the group treated with a free equivalent dosage of LMWF. Overall, strong evidence showed that, in our designed nanosystem, the release of LMWF upon NIR laser treatment would not damage normal epithelium cells, while the same treatment on cancerous SCC-9 cells caused significant cell death.

### 3.5. Detection of Cancer Cell Apoptosis

Apoptosis in cancer cells was detected using flow cytometry to further study the treatment effect of PCM-ICG-LMWF (+) upon laser irradiation. As shown in Figure 7C, the apoptotic rate of SCC-9 cells in the PCM-ICG-LMWF (+) and free equivalent dosage of LMWF groups increased significantly compared with the other groups. The results indicated that the LMWF released after laser irradiation, like the free LMWF, had a significant inhibitory effect on the proliferation of SSC-9 cells. The apoptosis rate was 44.44%, primarily because LMWF induced most SSC-9 cells to early apoptosis. As for the other groups, although some cells underwent apoptosis, most cells remained normal. For example, the apoptosis rate was 5.76%, indicating that those treatments had no obvious effect on the proliferation of SSC-9 cells. These results indicated that the release of LMWF from the patch upon NIR irradiation showed equivalent therapeutic effect compared with free LMWF. However, the on-demand release of LMWF endows us to kill the cancer cells at the expected treatment windows simply by using one patch without repeated administration, which will be more efficient for potential use in a real oral environment.

## 4. Conclusions

We report a novel multi-layered scaffold using electrospun nanofibers and electrospray particles for NIR-responsive on-demand release of LMWF, an anticancer drug, for squamous cell carcinoma therapy and recurrence prevention. In our designed system, LMWF is co-loaded with photothermal dye into PCM microparticles as a photothermal-gated material using coaxial electrospray techniques. The NIR-triggered release could still be detected after 5 cycles of laser irradiation, showing excellent photothermal stability and the properties of photothermal-responsive. In vitro experiments indicated that cell viability decreased to 5.06 ± 0.83% in the PCM-ICG-LMWF (+) group after three rounds of irradiation, and the apoptosis rate was 44.44%, while the control group was 5.76%. Thus, it is suggested that the controlled release of LMWF efficiently kills cancer cells. In addition, the top layer of the scaffold with strong adhesive abilities promotes HOK cell proliferation, which is essential for damaged zone healing. Due to its mechanical characteristics and hydrophobicity, the backing layer guarantees the in situ release of LMWF. In summary, this design and combination have great potential in treating and preventing the recurrence of oral cancer. To suit the use in an oral environment, such patches can easily stick to the damaged area because of the hydrophilicity of ADM in the inner layer. In particular, the triggered release of LWMF could be achieved via external NIR irradiation at the expected treatment window. In conclusion, we explore a novel way of mediation compared with conventional oral administration, which is in situ release, and the release could be controlled spatially. Additionally, the in situ release can show a direct therapeutic effect without blood circulation, compared to oral administration. Adding ADM could promote the proliferation of HOK cells lasting at least 5 days after killing cancer cells. Thus, such patches hold promising potential in killing cancer cells and simultaneously enhancing epithelium regeneration compared with direct drug intake.

## Data Availability

The data supporting the conclusions of this article are included within the article and its Appendix A. In addition, the data presented in this study are available from the corresponding authors on reasonable request.

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
