# Peer review of "On-Demand Release of Fucoidan from a Multilayered Nanofiber Patch for the Killing of Oral Squamous Cancer Cells and Promotion of Epithelial Regeneration"

_jfb, 2022, doi:10.3390/jfb13040167_

Round 1
Reviewer 1 Report (New Reviewer)
Regarding application multi-layered nanofiber patch capable of photothermal-triggered release, I am not getting how trigger site, which is oral cavity, light stimulus is correct justification? Do we have to shine light every time to initiate the release event in patient which underwent oral surgery? May be something salivary amylase-based trigger could be more fitting, can authors comment on this or may be add some discussion regarding method further to improve with correct trigger mechanism.
Abstract
Please change photothermal to photosensitive for the dye indocyanine green.
Introduction, sentence beginning with the ´Despite these improvements, it remains challenging to realize spatiotemporally controlled and on-demand release of the payloads to improve the treatment efficiency´ needs a suitable refence, cite a recent seminal work on the topic ttps://doi.org/10.3390/cells11182801 to update the recent development in spatiotemporal delivery of therapeutics as an in vitro model.
Materials methods
Can authors provide in table the optimization condition of electrospinning (ca. syringe diameter, flow rate, voltage, time, substrate coating distance etc.), similarly for the electro spraying as shown in figure 1.
Provide model, software version of all chemi-analytical instruments used in this study.
Results and discussion
Please provide diameter distribution and orientation of fibers quantitative data as shown with figure 2.C uniaxially random PCL nanofibers and 2.D random PCL/ADM nanofibers.
Provide catalogue number of all chemicals used in this study, majority info is missing as authors just provide suppler country/region only.
Figure 6. A Cell viability of various types of multi-layered scaffolds toward HOK cell lines, there is no difference between positive and negative control (unexposed vs. exposed), however, looking at the quantitative viability plot, there seems difference in (+) vs. (-) population, can authors comment on that?
Further, for quantifications in fluorescent based readouts authors implemented, ICG being fluorescent in nature might had leaked into absorption/emission spectrum of other fluorescent markers, please mention in detail the fluorescent markers selection based in ICG emission/absorption spectral range.
What gating criteria authors chose while analysis of FACS data to quantify apoptotic population scatter plot in figure 7.C.
I find very superficial discussion of the result in results and discussion section, this needs to be substantially strengthened for the manuscript.
In section 3.4. Evaluation of the treatment effect on HOK and SCC-9 cells, cite a latest https://doi.org/10.1021/acs.langmuir.2c00671 report on the method development shown with the opening sentence starting with ´To demonstrate therapeutic results in HOK and SCC-9 cells´ to update the references.
Provide a list of all abbreviations as some of them are used without defining in the main draft, which block the flow of discussion.

Author Response
Point-by-point response to review comments:
Reviewer #1: Regarding application multi-layered nanofiber patch capable of photothermal-triggered release, I am not getting how trigger site, which is oral cavity, light stimulus is correct justification? Do we have to shine light every time to initiate the release event in patient which underwent oral surgery? May be something salivary amylase-based trigger could be more fitting, can authors comment on this or may be add some discussion regarding method further to improve with correct trigger mechanism.
Response: We are very grateful for the reviewer’s comments on the recognition of our work and we have made the following amendments to the revised manuscript in accordance with the comments.
In fact, we need to initiate release every time. The reason why we select the NIR light to trigger the release is that NIR has high spatial and temporal resolutions and no toxicity to cells and tissues. Moreover, in that case, we aim to explore a novel way of mediation compared with conventional oral administration, which is in situ release, and the release could be controlled spatially without blood circulation like oral administration. Thus, we hypothesized that this design can show a direct therapeutic effect.
We agree with your opinion and suggestion. Now, we are seeking a better way to trigger release during the process of designing the patch. We believe that the nanofibers patch based on a salivary amylase-based trigger could be improved in our future work. Thank you for your suggestion honestly.
Comment: Abstract: Please change photothermal to photosensitive for the dye indocyanine green.
Response: Thanks for your suggestion. We have changed the word ‘photothermal’ to ‘photosensitive’.
Comment: Introduction, sentence beginning with the ‘Despite these improvements, it remains challenging to realize spatiotemporally controlled and on-demand release of the payloads to improve the treatment efficiency’ needs a suitable reference, cite a recent seminal work on the topic ttps://doi.org/10.3390/cells11182801 to update the recent development in spatiotemporal delivery of therapeutics as an in vitro model.
Response: Thanks for your suggestion and assistance in improving the quality of the Introduction. The literature (Ref 19) is cited on page 2.
Despite these improvements, it remains challenging to realize spatiotemporally controlled and on-demand release of the payloads to improve the treatment efficiency [19].
Comment: Materials methods: Can authors provide in table the optimization condition of electrospinning (ca. syringe diameter, flow rate, voltage, time, substrate coating distance etc.), similarly for the electro spraying as shown in figure 1.
Response: Thanks for your constructive comment. The table has been supplemented in section 2.2 on page 5 in the revised manuscript.
Table 1. The parameter of fabricating electrospinning nanofibers and electrospray nanoparticles.
|
Syringe diameter (mL) |
Gauge |
Voltage (kV) |
Flow rate (mL·h-1) |
Time (min) |
Distance (cm) |
PCL |
5 |
22 |
+15/-3 |
2.0 |
30 |
15 |
PCL/ADM |
5 |
22 |
+20/-3 |
2.0 |
60 |
15 |
PCM |
5 |
19 |
+15/-3 |
3.0 (outer) 1.0 (inner) |
10 |
15 |
Provide model, software version of all chemi-analytical instruments used in this study.
Response: Thanks for this valuable comment. We apologize for the careless behavior. The model and software version of all chemi-analytical instruments are in section 2.7 Characterization and section 2.8 Statistical analysis on page 7.
2.7. Characterization
The diameter, size distributions, and morphology of the material were observed by scanning electron microscopy (SEM, SUPRATM 55, Zeiss, Berlin, Germany). The NIR was produced by NIR laser (HW808AD, Hongwaixian, ShenZhen, China). The temperature change was recorded by thermal infrared imager (Testo869, Detu, Shanghai, China). The release profiles of RB, BSA and LMWF released from PCM were measured by Micro UV-vis spectrophotometer (NanoDrop™ One, Thermo Fisher, USA). The electrospun nanofiber was produced by electrospinning equipment (ET-2535H, Ucalery Technology, Beijing, China). The fluorescence images of PCM nanoparticles loaded with RB were observed by fluorescence microscope (DP74, Olympus, Tokyo, Japan). The cell morphology of HOK and SCC-9 were observed by fluorescence microscope (Nikon DS-Ri2 Zoom, Nikon, Tokyo, Japan). The optical images were obtained by digital cameras (Nikon D7500, Nikon, Tokyo, Japan). The apoptosis data were obtained by a flow cytometer (CytoFLEX, Beckman Coulter, Fullerton, California, USA).
2.8. Statistical analysis.
The multiple comparison procedures between different groups were carried out by one-way ANOVA. Each group was repeated at least three times and the statistical data was presented in a way of mean ± standard deviation. The significant difference was determined if P<0.05. The statistical analysis on the temperature change, in vitro cargos release, and cell viability as implemented by origin 2018. The size of nanofiber diameter and nanoparticles were measured by Image J. The apoptosis data from flow cytometry were recorded and processed by software-CytExpert for DxFLEX 2.0. The fluorescence images of PCM nanoparticles loaded with RB were recorded and processed by software called cellsens Dimension. The cell morphology of HOK and SCC-9 were recorded and processed by software called NIS-ELEMENTS D.
Comment: Results and discussion: Please provide diameter distribution and orientation of fibers quantitative data as shown in Figure 2.C uniaxially random PCL nanofibers and 2.D random PCL/ADM nanofibers.
Response: Thank you for your valuable suggestion. The quantitative data of diameter distribution on PCL, PCL/ADM nanofibers, and PCM nanoparticles have been added as supporting information. And some descriptions have been added in section 3.1 on page 9.
Figure S1 The diameter distribution of (A) PCM nanoparticles, (B) PCL nanofibers, and (C) PCL/ADM nanofibers.
From Figure S1B, C, we could observe that the diameter distribution of PCL and PCL/ADM nanofibers all showed a relatively centralized distribution at about 500-600 nm. The PCM microparticles between the two layers were fixed by the coaxial electrospray technique. The fluorescence photograph in Figure 2A was used to determine the payloads encapsulated in PCM particles. The SEM image in Figure 2B shows the distributed microparticles. Although these particles were not ideally uniform in size (589.55±163.94 nm), the size distribution was concentrated on 500-600 nm (Figure S1A).
Comment: Provide catalogue number of all chemicals used in this study, majority info is missing as authors just provide suppler country/region only.
Response: Thanks for this valuable comment. We apologize for the careless behavior. The catalog number of all chemicals has been listed in section 2.1 materials and reagents on page 3.
Low molecular weight LMWF (Mw≈8177 Da) was extracted and provided by the Institute of Oceanology, Chinese Academy of Sciences (Qingdao, China). ICG (Catalog no. YZ-1340009) and 4',6-diamidino-2-phenylindole (DAPI) (Catalog no. S2110) were purchased from Solarbio (Beijing, China). Rhodamine-B (Catalog no. 83689) was purchased from Sigma-Aldrich (America). Bull serum albumin (BSA) (Catalog no. BS114) was obtained from Biosharp (Anhui, China). Lauric acid (97%) (Catalog no. L0011) and stearic acid (95%) (Catalog no. S0163) were obtained from Tokyo Chemical Industry (Japan). Cell counting kit-8 (CCK-8) (Catalog no. C6050) was purchased from NCM Biotech (Suzhou, China). PCL (Mw≈80000 g mol−1) (Catalog no. 440744) was obtained from Sigma-Aldrich (USA). Phalloidin-iFluor 488 (Catalog no. ab176753) and Phalloidin-iFluor 555 (Catalog no. ab176756) were purchased from Abcam (Shanghai, China). Dulbecco's modified eagle medium (DMEM) (Catalog no. 12100) and antibiotic-antimycotic (Catalog no. P1400) were purchased from Solarbio (Beijing, China). Fetal bovine serum (FBS) (Catalog no. P30-3302) was purchased from Pan (Germany). Trifluoroethanol (TFE) (Catalog no. T63002) was obtained from Sigma-Aldrich (USA). Human OSCC cell lines SCC-9 and HOK cell lines were reanimated from the cryopreserved cells provided by Qingdao Stomatological Hospital (Qingdao, China). Annexin V-APC/PI apoptosis detection kit (Catalog no. E-CK-A217) was bought from Elabscience Biotechnology (Wuhan, China).
Comment: Figure 6. A Cell viability of various types of multi-layered scaffolds toward HOK cell lines, there is no difference between positive and negative control (unexposed vs. exposed), however, looking at the quantitative viability plot, there seems difference in (+) vs. (-) population, can authors comment on that?
Response: Thanks for your comment. The differences marked in the plot were attributed to the cells in the two groups cultured on well rather than the nanofibers with ADM. Thus, the cells could not show a similar proliferation extent compared with other groups due to the lack of proliferative effect of ADM. We do the description in section 3.4 on page 14 in the original submitted manuscript.
As shown in Figure 6A, HOK cells showed the same trend compared with the single PCM microparticles, except for the control and free equivalent dosage of LMWF groups. Although the absorption of the two groups was lower than that of the other groups, this could be explained by the addition of ADM, which promoted cell proliferation in the other groups.
Comment: Further, for quantifications in fluorescent based readouts authors implemented, ICG being fluorescent in nature might had leaked into absorption/emission spectrum of other fluorescent markers, please mention in detail the fluorescent markers selection based in ICG emission/absorption spectral range.
Response: We are sorry that the fluorescence images (as shown in Figure 6B and Figure 7B) are only used in observing the morphologies of cells, not quantification. The quantitative data (Figure 6A and Figure 6B) was cell viability obtained by CCK-8. In addition, the fluorescence of ICG was excited at 800 nm which has no effect on fluorescent markers[1].
Comment: What gating criteria authors chose while analysis of FACS data to quantify apoptotic population scatter plot in figure 7.C.
Response: Thanks for your comment. We used Annexin-V-APC/PI Apoptosis Kit to detect the apoptosis rate. Annexin V is a calcium-dependent phospholipid-binding protein with a high affinity for phosphatidylserine (PS), which can bind to the cytoplasmic membrane of early apoptotic cells through extracellularly exposed PS. Annexin V-APC fluorescence assay was used to detect apoptosis by flow cytometry. PI can specifically bind to double-stranded DNA and generate strong fluorescence. PI can enter cells to stain DNA due to loss of membrane integrity in late apoptotic or necrotic cells and when used in conjunction with Annexin V, can differentiate cells in different stages of apoptosis.
In Fig.7C, Annexin V-APC positive fluorescence is 105-107 on the horizontal coordinates, i.e., APC-A 105-107 represents the apoptotic cells. In addition, ECD-A represents PI, and ECD-A positive fluorescence is 105-107 on the vertical coordinates, i.e., ECD-A 105-107 represents the late apoptotic or necrotic cells. Therefore, the total APC-A 105-107 represents the apoptotic cells.
Comment: I find very superficial discussion of the result in results and discussion section, this needs to be substantially strengthened for the manuscript.
Response: Thanks for your constructive comment. The discussions have been strengthened, for example, section 3.1 on pages 8 and 9, section 3.3 on pages 11 and 12, and section 3.5 on page 18.
Our previous study proved that low-molecular high-sulfated polysaccharides exhibited cytotoxic toward SCC-9 cells while little cytotoxicity toward HOK cells [46]. In this study, we first extracted the LMWF and analyzed the composition. LMWF was extracted and produced according to the procedure in a previously published study. Briefly, dry algae were autoclaved in water. The hot solution, separated by continuous filtration, was concentrated and dialyzed. The fucoidan was precipitated and washed three times, then dried at 80°C. The monosaccharide composition of the LMWF was determined by a pre-column derivatization method. The molecular weight of fucoidan was measured and recorded using high-performance liquid chromatography. The composition analysis and the molecular weight are clearly displayed in Figure 1. The component we wanted was low-molecular high-sulfated polysaccharide, from Figure 1 we could see that the low-molecular high-sulfated polysaccharide including sulfate and fucose constitute the main composition, there were 36.85% and 35.07%, respectively. So we could conclude that the LMWF was extracted successfully as expected.
From Figure S1B, C, we could observe that the diameter distribution of PCL and PCL/ADM nanofibers all showed a relatively centralized distribution at about 500−600 nm. The PCM microparticles between the two layers were fixed by the coaxial electrospray technique. The fluorescence photograph in Figure 2A was used to determine the payloads encapsulated in PCM particles. The SEM image in Figure 2B shows the distributed microparticles. Although these particles were not ideally uniform in size (589.55±163.94 nm), and the size distribution was concentrated on 500-600 nm (Figure S1A).
In comparison, when RB-loaded PCM microparticles were heated to 40°C, 96.14±0.15% (n=3) of the RB was released from the microparticles. Thus, it was suggested that the PCM exhibited sustained release behavior once the temperature was the higher melting point. The release behavior was also studied under heating/cooling cycles to further examine the thermal-responsive behavior. As shown in Figure 4D, RB was released rapidly during heating, but the cumulative release changed very little during the cooling stage. Combining with the results from Figure 4C, we could conclude that the release behavior was temperature sensitive, once the temperature exceeded the PCM melting point, the cargo could be released from microparticles easily. When the ambient temperature was lower than the PCM melting point, hardly any cargos were released from nanoparticles. To summarize, the PCM nanoparticles were controlled-release platform depended on temperature.
Apoptosis in cancer cells was detected using flow cytometry to further study the treatment effect of PCM-ICG-LMWF (+) upon laser irradiation. As shown in Figure 7C, the apoptotic rate of SCC-9 cells in the PCM-ICG-LMWF (+) and free equivalent dosage of LMWF groups increased significantly compared with the other groups. The results indicated that the LMWF released after laser irradiation, like the free LMWF, had a significant inhibitory effect on the proliferation of SSC-9 cells and the apoptosis rate was 44.44%, primarily because LMWF induced most SSC-9 cells to early apoptosis. As for the other groups, although some cells underwent apoptosis, most cells remained in a normal state, for example, the apoptosis rate was 5.76%, indicating that those treatments had no obvious effect on the proliferation of SSC-9 cells. These results indicated that the release of LMWF from the patch upon NIR irradiation showed equivalent therapeutic effect compared with free LMWF. However, the on-demand release of LMWF endows us to kill the cancer cells at the expected treatment windows simply by using one patch without repeated administration, which will be more efficient for potential use in a real oral environment.
Comment: In section 3.4. Evaluation of the treatment effect on HOK and SCC-9 cells, cite a latest https://doi.org/10.1021/acs.langmuir.2c00671 report on the method development shown with the opening sentence starting with ´To demonstrate therapeutic results in HOK and SCC-9 cells´ to update the references.
Response: Thanks for your suggestion. We have added the latest report in section 3.4 on page 14.
To demonstrate therapeutic results in HOK and SCC-9 cells, we quantified the cell viability of these two cell types cultured on PCM microparticles using the CCK-8 method [47].
Provide a list of all abbreviations as some of them are used without defining in the main draft, which block the flow of discussion.
Response: Thank your valuable comment. We supplemented a list of abbreviations at the section abbreviation at the end of the article.
Abbreviations
OSCC: Oral squamous cell carcinoma; LMWF: low molecular weight fucoidan; PCMs: phase-change materials; ICG: indocyanine green; NIR: near-infrared; PCL: polycaprolactone; ADM: acellular dermal matrix; HOK: human oral keratinocytes; DAPI: 4',6-diamidino-2-phenylindole; BSA: Bull serum albumin; CCK-8: Cell counting kit-8; DMEM: Dulbecco's modified eagle medium; FBS: Fetal bovine serum; TFE: Trifluoroethanol; PBS: phosphate-buffered saline.

Reviewer 2 Report (New Reviewer)
The article "On-demand release of fucoidan from a multilayered nanofiber patch for the killing of oral squamous cancer cells and promotion of epithelial regeneration" describes a composite material capable to release under NIR photothermal stimuli the loaded drug (fucoidan) to treat oral cancer.. It is a valuable study that can be published after authors address the following problems:
The English language needs some minor polishing.
All abbreviations must be explained at first use (PBS row 176, explanation phosphate-buffered saline I suppose).
In introduction a stronger recent literature survey is necessary especially on antitumoral patches, nanoparticles and fibres. The author need to update the introduction by citing following doi: 10.2174/138920101602150112151157; doi: 10.3390/pharmaceutics13070957; doi: 10.3390/nano12111943; This will help the authors relate their findings to other research results.
In figure 3 the legend is obscuring the top part of the graphs. All graphs should start at time 0 (also in figure 4). Please correct it.
How is this system a better one? Conclusion section must be reworked to underline the novelty and advantages of this research, with actual numbers.
Author Response
Point-by-point response to review comments:
Reviewer #2: The article "On-demand release of fucoidan from a multilayered nanofiber patch for the killing of oral squamous cancer cells and promotion of epithelial regeneration" describes a composite material capable to release under NIR photothermal stimuli the loaded drug (fucoidan) to treat oral cancer. It is a valuable study that can be published after authors address the following problems:
Response: We are very grateful for the Reviewer’s comments on the recognition of our work.
Comment 1: The English language needs some minor polishing.
Response: Thanks for your constructive comment. We polished the English language, proofread the entire text carefully, and corrected English mistakes.
Comment 2: All abbreviations must be explained at first use (PBS row 176, explanation phosphate-buffered saline I suppose).
Response: Thanks for your constructive comment. We have explained the abbreviations at first use. And we also added a list of abbreviations in the section abbreviation at the end of the article. We apologize for our careless behavior again.
Comment 3: In introduction a stronger recent literature survey is necessary especially on antitumoral patches, nanoparticles and fibres. The author need to update the introduction by citing following doi: 10.2174/138920101602150112151157; doi: 10.3390/pharmaceutics13070957; doi: 10.3390/nano12111943; This will help the authors relate their findings to other research results.
Response: Thanks for your valuable comment. The literatures are cited in the section introduction on page 2.
To date, many implantable and degradable biomaterials have exhibited the capabilities of killing cancer cells and inhibiting tumor recurrence due to enhanced targeting and specificity [7-9].
Comment 4: In figure 3 the legend is obscuring the top part of the graphs. All graphs should start at time 0 (also in figure 4). Please correct it.
Response: Thanks for your constructive comment. We have adjusted the position of the legend. as for the second question, in fact, all graphs started at time 0, we don’t want the axis to shield the error bar at time 0, so we don’t start at time 0. So we hope that you can forgive this matter.
Figure 3. Temperature change curve of PCM microparticles loaded with different concentrations of ICG (0, 0.1, 0.2, 0.4, 0.8 and 1.0 mg/mL) placed A in air and B in PBS under the same irradiation density of 2200 mW. Temperature change curve of PCM microparticles loaded with ICG (1.0 mg/mL) under various irradiation densities (1800, 2200, 2600, 3000, and 3400 mW) placed C in air and D in PBS.
Comment 5: How is this system a better one? Conclusion section must be reworked to underline the novelty and advantages of this research, with actual numbers.
Response: Thanks for your valuable comment. The conclusion has been reworked in section conclusion on page 18.
We report a novel multi-layered scaffold using electrospun nanofibers and electrospray particles for NIR-responsive on-demand release of LMWF, an anticancer drug, for squamous cell carcinoma therapy and recurrence prevention. In our designed system, LMWF is co-loaded with photothermal dye into PCM microparticles as a photothermal-gated material using coaxial electrospray techniques. The NIR-triggered release could be still detected after 5 cycles of laser irradiation, which shows excellent photothermal stability and photothermal-responsive release property. In vitro experiments indicated that cell viability decreased to 5.06±0.83% in the PCM-ICG-LMWF (+) group after three rounds of irradiation and the apoptosis rate is 44.44% while the control group is 5.76%. Thus, it is suggested that the controlled release of LMWF efficiently kills cancer cells. In addition, the top layer of the scaffold with strong adhesive abilities promotes HOK cell proliferation, which is essential for damaged zone healing. Due to its mechanical characteristics and hydrophobicity, the backing layer guarantees the in situ release of LMWF. In summary, this design and combination have great potential in treating and preventing the recurrence of oral cancer. To suit for the use in a real oral environment, such patches can easily stick to the damaged area because of the hydrophilicity of ADM in the inner layer. In particular, the triggered release of LWMF could be achieved via external NIR irradiation at the expected treatment window. In conclusion, we explore a novel way of mediation compared with conventional oral administration, which is in situ release and the release could be controlled spatially. And the in situ release can show a direct therapeutic effect without blood circulation like oral administration. The addition of ADM could promote the proliferation of HOK cells lasting at least 5 days after killing cancer cells. Thus, such class of patches hold promising potential in killing cancer cells and simultaneously enhancing epithelium regeneration when compared with direct drug intake.

Round 2
Reviewer 1 Report (New Reviewer)
The revised version of the manuscript is significantly improved after addressing the comments from the reviewer. The revised manuscript can be accepted in present form.
Author Response
We are thankful for the reviewers’ valuable time and constructive comments which help further improve the quality of our work, thanks again.
This manuscript is a resubmission of an earlier submission. The following is a list of the peer review reports and author responses from that submission.
Round 1
Reviewer 1 Report
The manuscript On demand release of fucoidan from a multilayered nanofiber patch for the killing of oral squamous cancer cells and promotion of epithelial regeneration is wel planned, nice written and falls within the scope of the journal. Authors present an approach to produce novel system for delivery of fucoidan, as an anticancer drug. In my opinion, after minor revision manuscript might be accepted for publication. My main concrns are listed below:
The title of the manuscript is overexpressed and should be rewritten to better present content fo the study.
More advantages of electrospun fibers should be presented in Intrduction section.
In the Introduction section the novelty of this study has to be clearly presented.
Dots should be removed at the end of the sections titles.
In vitro should be uniformly presented in italic in the whole manuscript.
Please explain how such a composition of LMWF might affect its anticancer properties.
In the manuscript Authors wrote about PCM microparticles using electrospun technique. As far as I know it is quite challenging to produce microparticles using electrospinning. Provide the size of the produced materials to avoid missunderstandings.
Section 3.2. There is a lack of proper explanations of the obtained phenomena, because simple result presentation is shown in this section. There is also a lack of proper discussion of the obtained results with previosly reported.
Lines 451-453. So 50-250 ug/mL concentration o LMWF was established in presented or previosly published study? Please clarify.
Authors worte that experiments were repeated at least three times,but in Figs. 3-5, tthere is a lack of error bars. Please improve.
References list required improvements to meet all of the journal standards.
In my opinion last sentence of the conclusions section is overestimated and should be written more realistic.
Reviewer 2 Report
The paper shows innovative work with demanding experimental design.
Minor concerns:
1. Figure 4A - x scale 600 s instead of 60
2. Several abbreviations occur in the text. The List of Abbreviations would help the understanding
3. The Authors should provide the average molecular weights of the applied model substances (e.g., nerve growth factors and indocyanine green (ICG); BSA)
4. The reason for the different release profiles in Figure 5 should be detailed.
Reviewer 3 Report
These authors proposed a multi-layered scaffold using electrospun nanofibers and electrospray particles for NIR-responsive on-demand release of LMWF, an anticancer drug, for squamous cell carcinoma therapy and recurrence prevention. The aim is too ambitious. Recurrence prevention?
In addition, the anti-cancer property of fucoidan has been demonstrated in vivo and in vitro in different types of cancers. Nevertheless, it has been rarely investigated for its anti-cancer properties in clinical trials.
The fucoidan structure and monosaccharide composition might vary depending on different factors such as the source of fucoidan, the time and location of harvesting and the extraction method, which can affect the fucoidan’s bioactivities. This fact was not investigated.
All rational of formulation requires a full explanation.
The function of PCL and ADM is unclear.
Mucoadhesive assessment should be better demonstrated. The mechanism should be elucidated.
The authors stated that" morphology and non-uniform distribution of PCM microparticles did not influence the loading efficiency and release of encapsulated substances". Results cannot conclude this fact. A shape and particle size-dependence study should be performed.
Results indicated that the release of LMWF from the patch upon NIR irradiation showed equivalent therapeutic effect compared with free LMWF. Thus, it is not understood the functions and applications of the patch. Which is the final dimension of the patch? How long should it be linked to oral mucosa? How is its elimination?
Safety studies without laser are lacking (same for healthy cells). Stability studies should be performed.
Reviewer 4 Report
The manuscript pharmaceutics-1861982 presents a multilayer scaffold using electrospun nanofibers and electrospray particles to release an anticancer drug in response to exposure to NIR irradiation. This work is relevant for the treatment of squamous cell carcinoma and the prevention of recurrence, but the presented manuscript is more like a technical report than a scientific manuscript. Section 3 mainly provides only a statement of the results obtained without a thoughtful analysis and conclusions. Section 3 is called Results and Discussion, and there is very little discussion in this section. In addition, I do not support publication of this manuscript for a number of the following reasons:
1. The authors refer to nanoparticles as a phase-change material. What method used by the authors showed the phase change?
2. Due to what intermolecular interactions was the hydrophobic layer of electrospun PCL nanofibers retained on the microparticles?
3. Are all the methods described in section 2 original and developed only within the framework of the submitted manuscript? If these techniques are borrowed from other literary sources, these sources should be indicated in Reference. And after the phrase "produced according to the procedure in a previously published study" it is necessary to give the reference number.
4. In section 2.3 the authors need to specify the characteristics of the laser source.
5. What solvents precipitated and washed three times the fucoidan?
6. In section 2 it is also necessary to provide a description of the experiment to determine the encapsulation efficiency. I also want to know the value of loading rhodamine B into the microparticles shown in Fig. 2A.
7. I do not understand the reason for the slight release of rhodamine B. How can the inner solution be encapsulated by the outer solution?
8. I ask the authors to explain what causes the binding and release of molecules?